materials science/nanotechnology

green tea, orange peel, gold nanoparticles, nanofibres, skin mask, transdermal delivery

**Author for correspondence:**
K. M. Nalin de Silva
e-mail: kmnd@chem.cmb.ac.lk

This article has been edited by the Royal Society of Chemistry, including the commissioning, peer review process and editorial aspects up to the point of acceptance.

# Nanofibrous cosmetic face mask for transdermal delivery of nano gold: synthesis, characterization, release and zebra fish employed toxicity studies

D. C. Manatunga[1], V. U. Godakanda[1],

H. M. L. P. B. Herath[1], Rohini M. de Silva[1],

Chen-Yu Yeh[2], Jiann-Yeu Chen[3],

A. A. Akshitha de Silva[2], S. Rajapaksha[4],

Renuka Nilmini[4] and K. M. Nalin de Silva[1]

[1]Centre for Advanced Materials and Devices, Department of Chemistry, University of Colombo, Colombo 00300, Sri Lanka
[2]Department of Chemistry, and [3]Research Centre for Sustainable Energy and Nanotechnology (RCSEN), National Chung Hsing University, Taichung 402, Taiwan
[4]Department of Engineering Technology, Faculty of Technology, University of Sri Jayawardenapura, Sri Lanka

DCM, 0000-0003-1489-5557; VUG, 0000-0002-2489-7903; RMdeS, 0000-0003-0955-6366; KMNdeS, 0000-0003-3219-3233

This study involves the generation of gold nanoparticles (Au NPs) via a novel natural/non-toxic methodology using tea and orange-peel extracts. These were then embedded into a novel blend composed of a polyethylene oxide and gelatin (PEO-Gel) fibre mat. The scanning electron microscopy results indicated that the addition of both collagen (COL) and ascorbic acid (AA) into the PEO-Gel system (PEO-Gel-AA-COL system) enhances the Au NP incorporation into nanofibres leading to a diameter of 164.60 ± 20.95 and 192.43 ± 39.14 nm in contrast to the spraying observed with the Au PEO-Gel system alone. Releasing studies conducted over 30 min indicated that the PEO-Gel-AA-COL-orange peel Au (OpAu) system accounts for a higher content of Au release than the green tea Au (GtAu) NP system where a maximum release could be attained within 10–30 min depending on the amount of Au NPs that have been incorporated. Moreover,

the transdermal diffusion studies conducted using Strat membrane indicated that Au NPs from both formulations (PEO-Gel-AA-COL-GtAu nanofibre, PEO-Gel-AA-COL-OpAu nanofibre) have diffused through the stratum corneum and trapped in the dermis and epidermis indicating its transdermal deliverability. Additionally, 2,2-diphenyl-1-picrylhydrazyl (DPPH) assay revealed that nanofibres have similar radical scavenging activity like AA standard. Toxicity evaluation on a zebra fish embryo model confirmed that both GtAu NPs and OpAu NPs do not induce any teratogenic activity and are safe to be used in the range of 1.0–167 µg ml$^{-1}$.

## 1. Introduction

As in every field, within the area of cosmetics, there is a huge diversification to improve the quality and to meet the demands of clients who prefer good and well-groomed personal care products. Among the vast number of cosmetic items, insertion of nanomaterials has given a better scope [1] and are widely used in sunscreen creams [2,3], anti-ageing creams [4,5], hair products [6,7], facial masks [8,9] also in lipsticks [10]. The cosmetic application of electrospun nanofibres is an emerging area where much attention has been devoted on developing facial masks or membranes capable of releasing skincare products [11,12].

Electrospinning is a simple, low cost, versatile technique [13,14] of obtaining nanofibres for different cosmetic applications with a diameter in the micrometre to nanometre range using polymer solutions such as cellulose acetate [15], chitosan [16,17], hyaluronic acid [18,19] as well as synthetic polymers, namely, polyvinyl alcohol (PVA) [9], polyvinyl pyrrolidone (PVP) [20], and polyethylene oxide (PEO) [21]. The main advantage of the electrospinning technique is its ability to incorporate any active compounds of interest while fabricating the fibrous mat. Further, electrospun fibre masks require no preservatives, and are efficient in releasing active compounds and they could also be packed as dry sheets [22]. In order to release active compounds, they need to be wetted only at the required time [23]. Owing to their ability to allow better contact with the skin aiding a deeper penetration of the active agent [11], these have been mainly targeted to release compounds for skin healing [24], therapy [25] and cleansing [26].

The importance of gold nanoparticles (Au NPs) in skin care has been identified over the years as they are known to reduce the wrinkling of the skin [27], improve brightening, provide anti-bacterial, anti-inflammatory, skin healing, and a cleansing effect while slowing down the collagen depletion [28] and elastin breakdown [29,30]. Nevertheless, most of the applications of Au NPs have been carried out in the field of drug delivery [31–34], heavy metal removal [35] and as biosensors of Au nanofibres with various polymers [36–42]. Skin care products such as hydrogels [29], conventional Au facial masks [43], night cream, eye cream, whitening cream [28,44], using bulk and nano Au are reported [45]. After careful analysis of the reported data we found not much work on the electrospun Au nanofibres for skin care application. Work conducted by Fathi-Azarbayjani and colleagues [9], have used PVA, cyclodextrin to develop a nanofibre mask for the topical delivery of retinoic acid, ascorbic acid (AA), collagen (COL) and Au NPs that has been synthesized chemically.

Therefore, in our work we thought of developing Au NPs using natural plant extract and to integrate this with the electrospun fibremat to obtain an extremely biocompatible face mask for beauty treatments. For this purpose, we designed a nanofibrous facial mask containing green tea extract and orange-peel extract derived Au NPs (GtAu NPs and OpAu NPs respectively) blended with biocompatible polymers of PEO and gelatin via an electrospinning technique. PEO is a cheap synthetic amphiphilic block co-polymer used in a wide range of applications and in cosmetic formulations mainly owing to its hydrophilic, biocompatible, biodegradable and non-toxic behaviour [21,46]. On the other hand, gelatin, which has widely been used as a gelling agent in drugs, food and cosmetics, is a natural, elastic, biocompatible polymer that exhibits a hydrophilic, biocompatible and biodegradable nature [47–50]. However, the electrospinning of gelatin is reported to be difficult [47] and therefore, gelatin is generally mixed with high molecular weight polymers like PVA [51], polycaprolactone (PCL) [52] or PEO [47] to improve the spinnability and physico-chemical properties of gelatin [53].

In general, cosmetic facial masks also contain brightening agents, vitamins, moisturizers, minerals, protein and herbal ingredients, out of which COL is used as a skin nutrient to reduce skin wrinkling while L-AA is to reduce pigmentation [43]. COL contains a higher surface charge which makes it harder for electrospinning [54]. This has been overcome by blending collagen with polymers such as PEO [54]. Therefore, ability of incorporation of COL and AA to develop a complete beauty care facial mask was aimed at in our study. Hence the final objective of this study can be stated as producing a dry mask, containing a novel blend of Au NPs, COL, AA, PEO gelatin (PEO-Gel), via a water-based

electrospinning technique which shows the ability of releasing ingredients by simple wetting. Green synthesized Au NPs were obtained using tea and orange-peel extracts and these were later blended with a PEO-Gel, COL and AA mixture to obtain a nanofibrous system. The developed facial mask was subjected to release studies and transdermal diffusion studies to analyse the Au NP release and skin permeation. Furthermore, the antioxidant property and toxicity evaluation of the Au NPs were conducted using DPPH assay and zebra embryo model-based toxicity studies, respectively.

# 2. Material and methods

$HAuCl_4 . 3H_2O$, PEO ($100\,000$ gmol$^{-1}$), gelatin (Type B, gel strength 225 g), L-AA, and COL were purchased from SIGMA Aldrich. Green tea leaves were purchased from Watawala Tea Ceylon Ltd, whereas orange peel was collected from fresh oranges and dried under a dark environment. Double-distilled water (DDW) was used throughout the experiment.

## 2.1. Green tea extract and orange peel extracted mediated synthesis of gold nanoparticles and characterization

Au NPs were prepared by adapting an already reported procedure with some modifications. Green tea extract was prepared by boiling tea leaves for 15 min in DDW. The obtained yellow orange extract was centrifuged at 3500 r.p.m. for 10 min and filtered through the whatmann paper (110 mm) to obtain a concentrated 1.0% (w/v) tea extract solution. This was then diluted to obtain 0.1% working solution to react with the Au precursor. To a 3.0 ml of 1 mM Au(III) solution, 1.6 ml (optimized value obtained) of the 0.1% green tea extract was added and stirred vigorously (800 r.p.m.) at 40°C. After a red wine colour is developed the reaction was allowed to take place at room temperature and provided with vigorous stirring (1100 rpm). At the end of 15 min the obtained red wine colour solution was centrifuged at 3500 r.p.m. for 20 min followed with a washing step to remove unreacted constituents. It was then redispersed in DDW to obtain a 10 mg/7.5 ml GtAu NPs working solution for electrospinning purposes.

Similarly, the dried orange peel was cut into small pieces and boiled in DDW for 15 min to obtain 10% (w/v) orange peel extract. This was then centrifuged (3500 rpm for 10 min), filtered and later pH was adjusted to 7. From this extract 1.0 ml was added in to 2.0 ml of Au (III) solution kept at 40°C. This was vigorously stirred at 800 rpm until the red wine colour is obtained, later on the temperature of the system was reduced to room temperature and the stirring was continued at 1100 r.p.m. for 15 min. The end product was a similar procedure as in GtAu NPs to obtain 10 mg/7.5 ml OpAu NPs working solution for electrospinning purposes.

As synthesized, GtAu NPs and OpAu NPs were characterized using ultraviolet–visible spectrophotometry (UV–Vis) measurements (Carry 300, USA) to identify the characteristic plasmonic bands [50]. The shape of the NPs was observed by transmission electron microscopy (TEM). Samples of the AU NPs were prepared by placing drops of the product solution onto carbon-coated copper grids and allowing the solvent to evaporate. TEM measurements were performed on a JEOL model JEM-2010 and JEM- 2100 instrument models (Tokyo, Japan) operated at an accelerating voltage of 200 kV. Fourier transform infrared (FTIR) spectra were recorded (Perkin Elmer Spectrum 100) in the range of 400–4000 cm$^{-1}$.

## 2.2. Preparation of electrospinning solutions

In this study, PEO was used as the base material into which gelatin and other compounds were added to obtain the electrospinnable solution. In order to spin PEO individually, 4.5% (w/v) of PEO viscous solution was prepared in DDW. To prepare PEO-Gel blends, PEO and gelatin was mixed in 9:1 ratio with a total polymer weight of 4.5%. Subsequently to prepare the PEO-Gel-AA-COL blend 27.7% (w/w) of PEO, 3.08% (w/w) of Gel, 6.84% (w/w) of AA and 0.68% (w/w) of COL were mixed together to obtain a total polymer weight of 4.5%.

Likewise to prepare the Au NPs incorporated nanofibres, 0.5 mg, 0.75 mg and 1.0 mg of Au NPs from the two systems (GtAu NPs and OpAu NPs) were first mixed with PEO-Gel blend having 9:1 PEO:Gel in 4.5% (w/v) polymer weight. Similarly, 0.5 mg, 0.75 mg and 1.0 mg of Au NPs were also mixed with PEO-Gel-AA-COL optimized blend having total polymer weight of 9.65% for GtAu NPs and 10.7% for OpAu NPs, respectively (total polymer weights were obtained based on the spinnability studies conducted with different total polymer weight amounts). All solutions were stirred magnetically in closed bottles at room temperature for 12 h and the viscosity measurements

were obtained using a viscometer (LVT, Brookfield, USA). The conductivities were measured by a conductivity metre (Orion star A112, Thermoscientific, USA). Full details of the solutions prepared are listed in the electronic supplementary material, table S1.

## 2.3. Electrospinning of PEO-Gel, PEO-Gel-COL-AA and gold nanoparticles incorporated PEO-Gel, PEO-Gel-COL-AA and characterization

The respective solutions were loaded in to a 5 ml syringe (Terumo) fitted with a metal spinneret (20G) and mounted on a syringe pump (KDS100, Cole-Parmer, USA). The solution was expelled from the syringe at a rate of 0.3 ml h$^{-1}$ provided with a high voltage power supply (ZGF-2000, Shanghai Sute Electrical Co. Ltd., China) of 11.9–15.9 kV, in between the needle and a grounded collector covered with aluminium foil. The distance between the needle and collector was 13–18 cm depending on the system used for spinning. Electrospinning was carried out at a temperature of 21°C. After electrospinning for 6 h, the products were stored in a vacuum desiccator at room temperature for 24 h to remove residual moisture.

Fibre diameters were determined from scanning electron microscopy (SEM) images obtained using a SEM Carl Zeiss Evo 18 (Accel voltage: 10–20 kV, probe current: 1–25 pA). Moreover, the presence of Au NPs in the fibres were identified using the back scattering mode of SEM. FTIR spectra of the neat nanofibres and Au NPs incorporated nanofibres were taken with a Perkin Elmer (Spectrum 100) in the wavelength region 400–4000 cm$^{-1}$ using the attenuated transmission (ATR) mode with a 1 cm$^{-1}$ resolution and the signals are averaged from 32 scans. Powder X-ray diffraction (XRD) patterns of the produced fibres and the starting materials were obtained using a powder X-ray diffractometer (Rigako, SmartLab SE, Japan) using Cu K$\alpha$ radiation, with a step size of $2\theta = 5°$ min$^{-1}$ over the range of 2°–80°. The differential scanning calorimetry (DSC) thermograms of the nanofibres and the starting materials, were measured using a differential scanning calorimeter (SETARAM DSC 131 evo) by running the samples in the range of 20°C–350°C at 10°C min$^{-1}$ under a flow of nitrogen.

## 2.4. Release studies of gold nanoparticles via dialysis membrane

Release of Au NPs from best-performing Au NPs incorporated PEO-Gel nanofibres systems was assessed by incubating 30 mg of the Au nanofibres in 15 ml of phosphate saline buffer with (PBS) pH 7.4. In here the Au nanofibres were first entrapped inside a dialysis tubing (Thermo Snake Dialysis Tubing MWCO 7000 Da) which was then dipped in the receiving buffer solution. At specific time intervals (up to 30 min), 1.0 ml of the sample was withdrawn and replaced with fresh buffer and release of Au content was analysed via UV–Vis.

## 2.5. Skin permeation studies of gold nanoparticles

To assess the transdermal penetration of Au NPs through the skin, an artificial skin model (Merck Strat -M- membrane) was clamped between the donor and the receptor chamber of the Franz diffusion cell with an effective permeation area of 4.9 cm$^2$ (25 mm diameter) while having a receiver cell volume of 5.0 ml. PBS was used as both a donor and receiver solution while the donor compartment contained 53 mg of PEO-Gel-AA-COL-Au NPs. The whole set up was incubated at 37°C provided with stirring at 200 r.p.m. After 5 days of incubation, permeation of Au NPs through the skin layers was assessed by subjecting to cross-sectional SEM analysis and energy dispersive X-ray (EDX) analysis.

## 2.6. Free radical scavenging ability of the PEO-Gel-AA-COL-gold nanoparticle fibres

The potential antioxidant activity of the Au NP loaded fibres was determined based on the free radical scavenging activity of DPPH, with some modifications to the protocol described by Brand Williams *et al*. [55]; 200 µl aliquots of the samples with varying concentrations of Au NPs extracted from fibre pieces in ethanol were incubated with 1800 µl of 48 ppm DPPH solution prepared in ethanol. AA was used as a positive control. Absorbance values at 515 nm were determined after 10 min incubation in the dark. The per cent inhibition activity was calculated as follows:

$$\frac{(\text{absorbance of DPPH} - (\text{absorbance of the sample} - \text{absorbance of the control}))}{(\text{absorbance of DPPH})} \times 100,$$

where absorbance of sample is the absorbance of the DPPH solution with the released content from

nanofibres and absorbance of control is the absorbance of the released content from nanofibres in ethanol in the absence of DPPH.

## 2.7. Toxicity assessment via zebra fish embryo model

Sexually matured male and female zebra fish (*Danio rerio*) were sorted and prepared for breading and reared in mini-zebra fish facility at Biotechnology Laboratory, University of Colombo, as described elsewhere [52]. The 48 < hours post-fertilization (hpf) embryos were collected from overnight breeding rounds (in freshly prepared embryo (E3) medium) and the embryos sorted (for unfertilized or coagulated, damaged, opaque, asymmetric and no transparency) in Petri dishes, were washed twice or trice gently with distilled water while being carefully observed under an Olympus BX53 microscope (Olympus BX51, Japan). Subsequently, embryos were seeded in the wells of a 96 well plate in 20 µl of zebra fish culture medium. The embryos were exposed with different concentrations of GtAu NPs and OpAu NPs (1.67 ppm, 16.67 ppm, 33.3 ppm, 166.67 ppm, 208.32 ppm). Cd (II) solution with an effective concentration of 100 µg ml$^{-1}$ was used as a positive control where the wells with zebra fish culture medium served as the negative control. All the treatments were carried out in triplicate in approximately 28.5°C. Ultimately, developmental anomalies or teratogenicity parameters (pericardial edema, yolk sac edema, pericardial edema, bent trunk, tail malformation, and an uninflated swim bladder) were recorded at 24, 48, 72 hpf [53,54] under an Olympus BX53microscope (Olympus BX51, Japan) with a microscope digital camera (6.3MP Sony CMOS sensor).

# 3. Results and discussion

## 3.1. Synthesis of gold nanoparticles and characterization using ultraviolet–visible spectrophotometry and transmission electron microscopy analysis

In this study phytochemicals present in green tea extract and orange-peel extract were used for the reduction of Au precursor salt and to produce Au NPs. With the addition of the plant extracts to 1 mM HAuCl$_4$ solution, a colour change was observed from yellow to wine red/ruby red (figure 1$a$) indicating the formation of Au NPs. As given in figure 1$b$, the presence of the characteristic plasmon resonance band [56,57] centred at $\lambda_{max}$ of 527 nm for the GtAu NPs and 529 nm for the OpAu NPs confirmed the formation of the Au NPs. It is clear that the full width of half maximum (FWHM) of the plasmon peak of OpAu NPs is much wider than that of GtAu NPs, highlighting the increased particle size [58] of the OpAu NPs. A study on different amounts of reducing agents and different reaction temperatures was also carried out to select the optimum conditions for the preparation of the Au NPs (electronic supplementary material). The results (electronic supplementary material, figure S1) indicated that excessive amounts of the reducing agents and at high temperatures, the broadening and red shifting of the plasmonic band owing to the increase of particle size was as observed in previous studies [57,59].

The morphology and size of the synthesized Au NPs were determined using TEM and images obtained are shown in figure 1$c$,$d$. Particle sizes of GtAu NPs and OpAu NPs range from $18 \pm 3.5$ nm to $11.88 \pm 6.77$ nm, respectively. Broadening of the plasmon peak of OpAu NPs (figure 1$b$) could be owing to the clustering of smaller NPs leading to different shapes such as star or ellipsoidal creating agglomerates [60] as confirmed in TEM results (figure 1$d$).

## 3.2. Electrospinning of gold nanoparticles with PEO-Gel and PEO-Gel-AA-COL systems and morphological characterization

During the electrospinning process the total polymer weight of neat nanofibres (PEO, PEO-Gel) was maintained at 4.5% which was changed to 9.0% with the addition of AA and COL as additional constituents to prepare the facial mask. According to the results given in the electronic supplementary material, table S1, the presence of AA would have led to a reduction of the viscosity owing to the acid catalysed scissoring of PEO molecules [61]. By contrast, the addition of Gel alone has led to an increase of the viscosity of the PEO, as it acts as a gelling agent in aqueous medium [62]. However, the addition of Gel, AA and COL led to an increase in the conductivity of the PEO polymer solutions, and by combining with Au NPs it enhanced its electrospinnability [63]. The best results for electrospinning were achieved with 0.3 ml h$^{-1}$ flow rate, 18 cm distance between tip and collector, and 11.9 kV voltage for neat PEO nanofibres, whereas it was 0.3 ml h$^{-1}$, 18 cm, 18 kV for 4.5% wt PEO-Gel

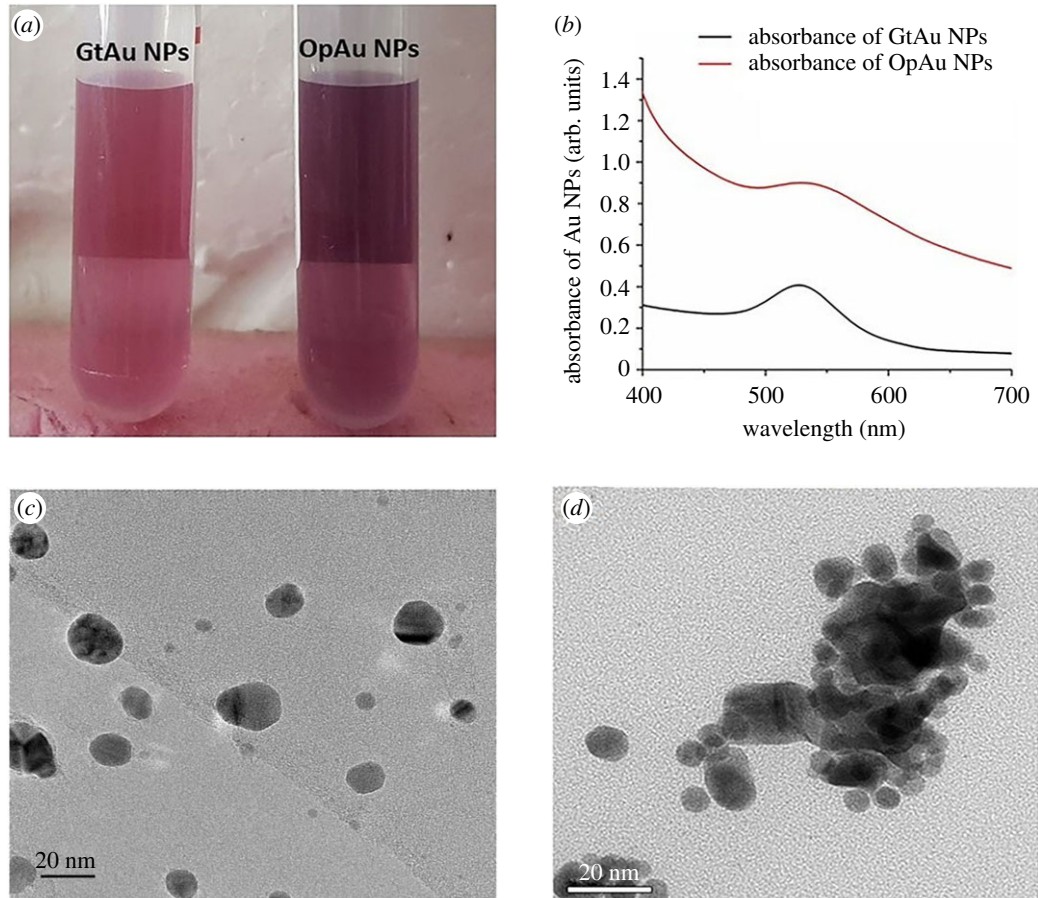

**Figure 1.** (*a*) Red wine colour Au NPs generated from green tea extract and orange peel extract, (*b*) UV visible spectrum of Au NPs generated from green tea (GtAu NPs) and orange peel (OpAu NPs) extracts, (*c*) TEM image of GtAu NPs, and (*d*) TEM image of OpAu NPs.

blend (electronic supplementary material, table S2). Similarly, when AA and COL were incorporated in to the PEO-Gel, spinning conditions were drastically changed as specified in the electronic supplementary material, table S1. The results of optimization studies were analysed by visual observations via optical microscope (data not shown). The images of the electrospinning apparatus and the resulting fibre mask are given in the electronic supplementary material, figure S2.

Morphological observation of neat nanofibres and Au NP incorporated nanofibres through SEM analysis (electronic supplementary material, figure S3a), indicated that PEO nanofibres were approximately smooth and defect-free with a mean diameter of 234.49 ± 21.49 nm with most of them falling in between 220 and 240 nm. As given in the electronic supplementary material, figure S3b, when gelatin Gel was introduced to the PEO polymer system, it led to a reduction of fibre diameter to an average size of 193.61 ± 21.46 nm giving a branching effect on the nanofibres. This reduction in diameter could be owing to the increase of the conductivity of the PEO-Gel system resulting from charged groups in the gelatin structure [64] that creates stronger elongating forces on the polymer jet, resulting in lowering of the fibre diameter [63]. Moreover, this effect was much more pronounced with further addition of Au NPs into PEO-Gel system as observed in the electronic supplementary material, figure S2c–d, owing to the increased conductivity [63]. The resulting nanofibres had a mean diameter of 178.93 ± 21.33 nm and 126.03 ± 22.99 nm for PEO-Gel-GtAu NPs and PEO-Gel-OpAu NPs respectively. However, the incorporation of GtAu NPs and OpAu NPs into PEO-Gel nanofibres are different from each other; GtAu NPs have been trapped in between the PEO-Gel nanofibres (electronic supplementary material, figure S3c) while OpAu NPs have been slightly embedded in to the PEO-Gel nanofibres (marked with a yellow arrow). As shown in the circled area of the electronic supplementary material, figure S3d, nanofibres of the PEO-Gel-OpAu NP system were merged together with the unspun polymer drops that were ejected occasionally owing to the higher viscosity of the solution (electronic supplementary material, table S1) which makes it difficult for solution flow and ejection [53].

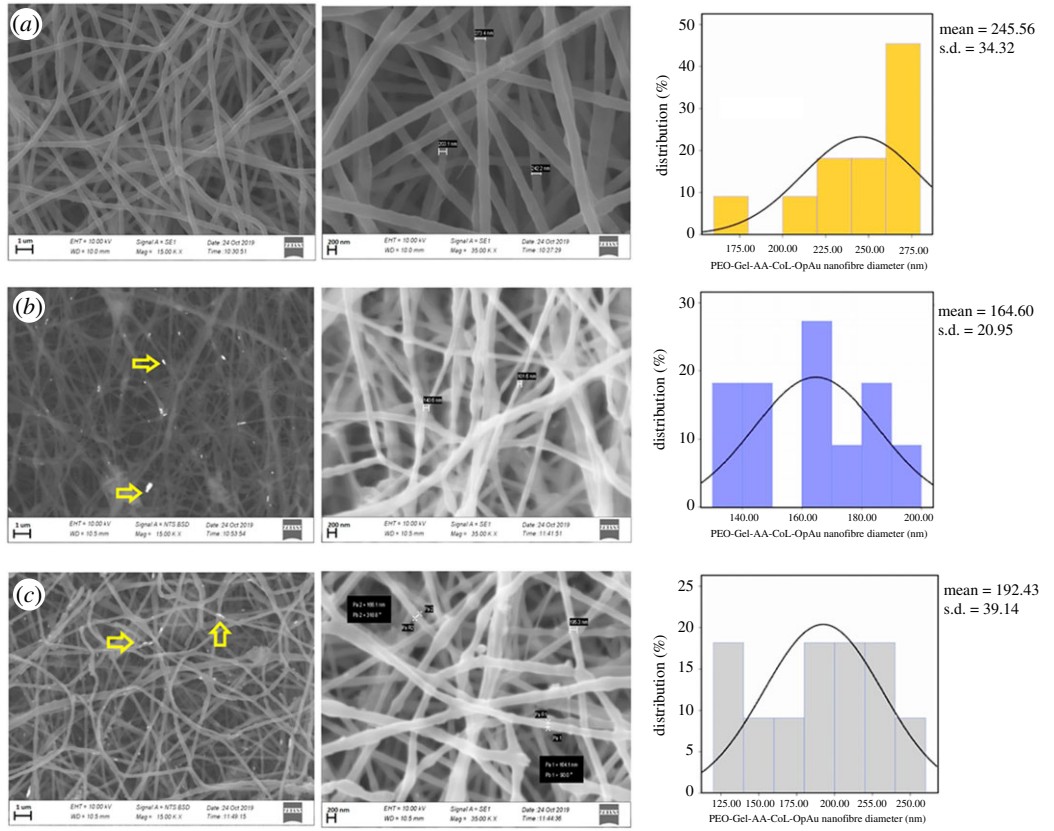

**Figure 2.** SEM images and size distribution histograms of (*a*) PEO-Gel-AA-COL nanofibre systems, (*b*) PEO-Gel-AA-COL-GtAu NPs, and (*c*) PEO-Gel-AA-COL-OpAu NPs (areas marked with arrows indicate the entrapped AuNPs inside the nanofibres).

By contrast, PEO-Gel-AA-COL systems were different from PEO-Gel nanofibre systems leading to less dripping and better Au NP incorporation into the nanofibre system (figure 2*a–c*). Nevertheless, the mean diameter of the PEO-Gel-AA-COL was much higher ($245.56 \pm 24.32$ nm) than that of neat PEO and PEO-Gel systems, which could be owing to the presence of many compounds accounting for higher total polymer weight. Moreover, the resulting fibres had a mean diameter of $164.60 \pm 20.95$ nm and $192.43 \pm 39.14$ nm for PEO-Gel-AA-COL-GtAu NPs and PEO-Gel-AA-COL-OpAu NPs respectively (figure 2*b,c*) Entrapment of the Au NPs within the fibres is very clear owing to the blobs/beads along the fibres. Using a backscattering mode these Au entrapped areas could be made visible as marked by arrows in figure 2*b,c*.

## 3.3. Fourier transform infrared characterization of PEO-Gel-AA-COL nanofibres

The presence of different functional groups on the GtAu NPs, OpAu NPs, PEO-Gel nanofibres, PEO-Gel-AA-COL and PEO-Gel-AA-COL-Au NPs nanofibres could be identified using FTIR spectroscopy. As given in the electronic supplementary material, figure S4, characteristic bands corresponding to asymmetric and symmetric vibrations of $–CH_2$ at 2949 cm$^{-1}$ and 2846 cm$^{-1}$ [65–67] were present in both GtAu NPs and OpAu NPs (electronic supplementary material, figure S4a and d) which were closely matched with the vibrational bands of green tea extract and orange-peel extract (electronic supplementary material, figure S4b,c) which highlighted that Au NPs were capped with plant derived molecules which might had reduced the Au (III) precursors and provide a stabilization effect [68]. Additionally, the reduction of –OH stretching intensity (3394 cm$^{-1}$) in coated Au NPs indicted the possible interaction of alcohols and phenols of plant extract with Au NPs [68]. In addition, the presence of bands at 1627 cm$^{-1}$ (–C = C– stretch), 1396 cm$^{-1}$(–C–N– amide I stretch), 1741 cm$^{-1}$ (–C–O–C– stretch) and 1037 cm$^{-1}$(–C–O– stretch) on top of Au NPs suggested the presence of aromatic compounds, proteins, polysaccharides and amino acids coming from the plant extracts [69]. The PEO-Gel nanofibre system, resembles the infrared spectrum of PEO, but it also contain peaks corresponding to gelatin that appears at 1640 cm$^{-1}$(amide I of –C = O stretching couples with –C–N

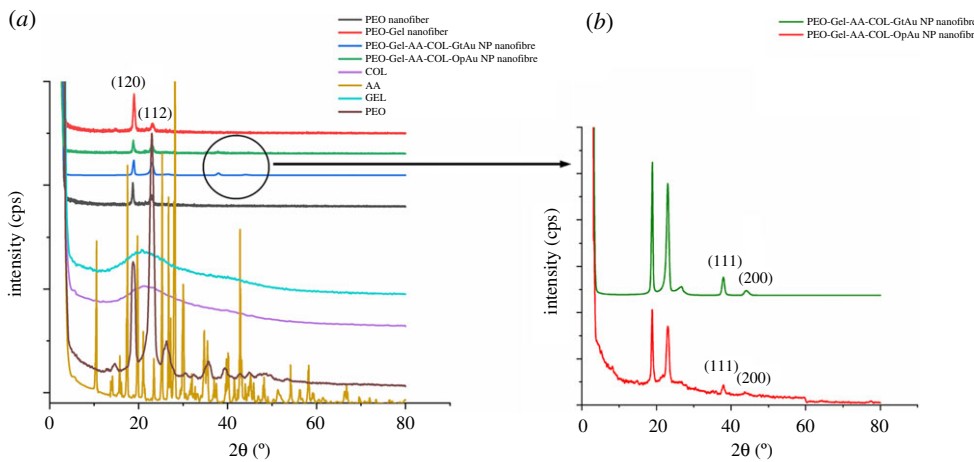

**Figure 3.** (*a*) XRD patterns of PEO, Gel, AA, COL, PEO nanofibre, PEO-Gel-AA-COL-GtAu nanofibre and PEO-Gel-AA-COL-OpAu nanofibre. (*b*) Enlarged image of PEO-Gel-AA-COL-GtAu nanofibre and PEO-Gel-AA-COL-OpAu nanofibre indicating the presence of peaks corresponding to Au NP.

stretch) and 1540 cm$^{-1}$ (amide II out of phase) [62]. A broader –OH stretching in the region of 3000–3500 cm$^{-1}$ could suggest an H-bond formation in between the two polymers [65,70]. Peaks at 1342 cm$^{-1}$ of –CH$_2$ symmetric wagging [46], 1466 cm$^{-1}$ of –OH bending, 1240 cm$^{-1}$, 1278 cm$^{-1}$ of –C–O– stretching [71], 1102 cm$^{-1}$ of –C–O stretching[46] and 2850 cm$^{-1}$, 2920 cm$^{-1}$ of symmetric and asymmetric stretching [72] suggests the main contribution is coming from PEO backbone. When Au NPs, COL and AA were blended with the PEO-Gel nanofibre system, there was no significant difference because the peaks of AA and COL were falling into the same region as PEO and Gel (electronic supplementary material, figure S4j,k).

## 3.4. Crystallographic characterization using X-ray diffraction analysis

Figure 3*a* displays the XRD patterns of PEO, Gel, AA, COL, PEO-Gel nanofibres, PEO-Gel-AA-COL-Au NP incorporated nanofibre systems. Both Gel and COL have a similar broad XRD patterns [73,74] highlighting their amorphous nature arising from the random coil conformation [74]. However, the incorporation of Gel and COL into PEO have not created much difference in the XRD pattern of respective fibre systems, which might arise owing to the abundance of the PEO molecules. Two sharp peaks were observed at 19.15° and 23.35° in PEO, PEO nanofibre, PEO-Gel and PEO-Gel-AA-COL nanofibre systems, corresponding to [120] and [112] facets that accounts for the semi crystalline nature in PEO [70,75]. When AA has been added to the PEO-Gel-AA-COL nanofibre system, the appearance of peaks corresponding to crystalline AA [76] was not visible, which could be owing to the lower content of AA used for the study which might have been hidden by the PEO prevalence. In PEO-Gel-AA-COL systems, peaks corresponding to [111] and [200] facets [77,78] of Au NPs were appearing at 2$\theta$ of 38.31°, 44.46° (figure 3*b*). Increased intensity at the 38.31° peak of Au NPs is indicative of the predominance of [111] facet during the formation of Au NPs [79].

## 3.5. Thermal analysis of PEO-Gel nanofibre systems

The effect of polymer blending and nanofibre formation on the thermal properties of PEO-Gel nanofibre systems was studied using DSC analysis. As given in figure 4, classic DSC curves were observed for PEO, Gel, AA and COL. In figure 4*a* and *e–i*, the melting point appearing around 74.3°C of neat PEO has shifted to lower values with lowered intensity when it was blended or formulated into nanofibres (72.3°C in PEO nanofibres, 73.1°C in PEO-Gel nanofibres, 72.8°C in PEO-Gel-AA-COL nanofibres and PEO-Gel-AA-COL-GtAu nanofibres, and 65.7°C in PEO-Gel-AA-COL-OpAu). This could be attributed to the alteration of PEO crystallinity during nanofibre formation [80].

Incorporation of Gel into PEO nanofibres, by obtaining an endothermic peak at 58.1°C (helix coil transformation of Gel) [81], broad peak around 75–100°C (water loss) [82] and final two endothermic peaks at 225°C (α-helix denaturation), 278°C (β-helix denaturation) [83] in neat Gel (figure 4*b*) were not visible in the nanofibres. Instead a plateau in specific regions (in addition to the broader

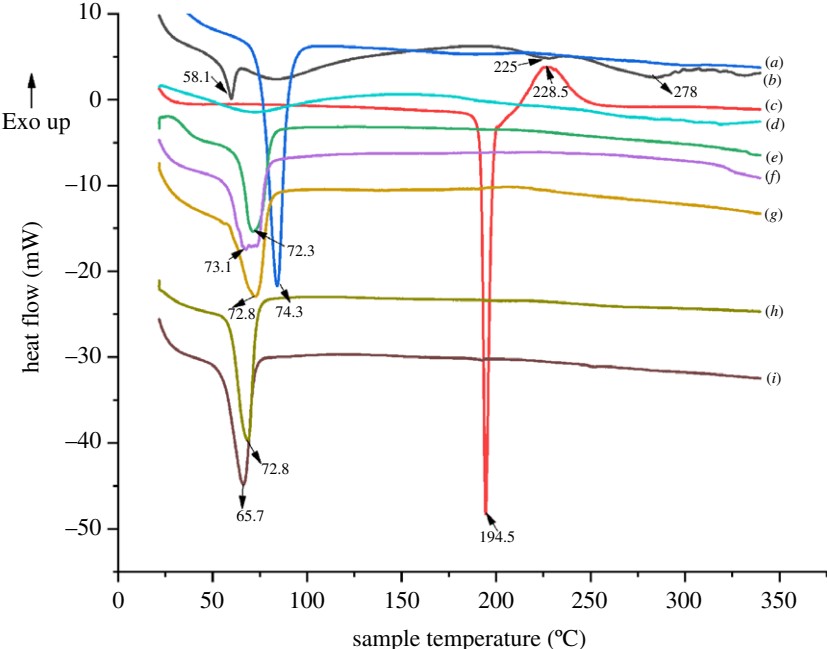

**Figure 4.** DSC thermograms of (*a*) PEO, (*b*) Gel, (*c*) AA, (*d*) COL, (*e*) PEO nanofibre, (*f*) PEO-Gel nanofibre, (*g*) PEO-Gel-AA-COL nanofibre, (*h*) PEO-Gel-AA-COL-GtAu nanofibre, and (*i*) PEO-Gel-AA-COL-OpAu nanofibre.

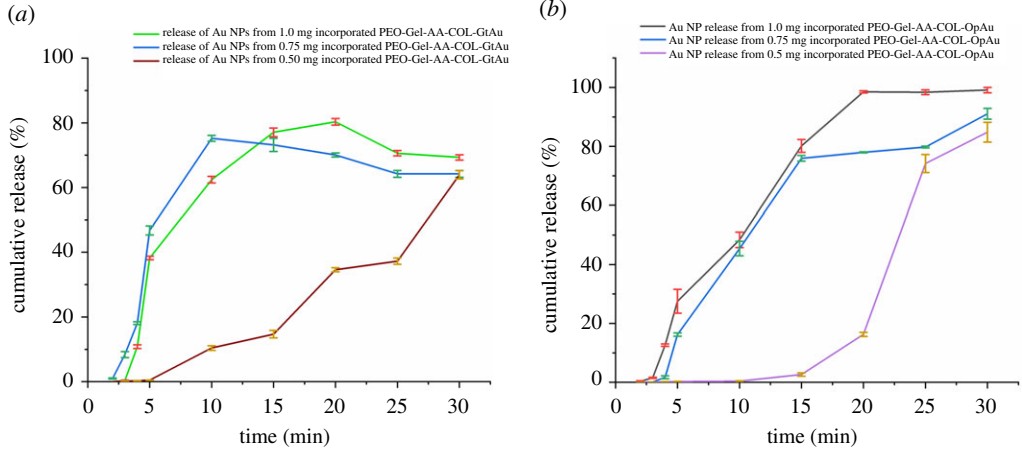

**Figure 5.** Release of Au NPs incorporated in three different levels (0.50 mg, 0.75 mg and 1.00 mg) from (*a*) a PEO-Gel-AA-COL-GtAu nanofibre system, and (*b*) a PEO-Gel-AA-COL-OpAu nanofibre system.

endothermic peak of PEO melting) was observed, indicating the reduction of crystallinity and alteration of the random coil structure of gelatin [62].

Similarly, a change in the thermal properties owing to the introduction of AA and COL in to PEO-Gel nanofibres was also observed which could be owing to the very low content of respective compounds added during the synthesis (figure 4*g*). However, as given in figure 4*h,i*, the increased nature of crystallinity was evident after integrating Au NPs into PEO-Gel-AA-COL systems [84].

## 3.6. Analysis of gold nanoparticles release from PEO-Gel-AA-COL systems

For the release studies of both GtAu and OpAu NPs, the PEO-Gel-AA-COL nanofibre system was selected out of other systems as it has shown a proper and a stable entrapment of Au NPs inside the respective nanofibres (figure 2*b,c*). These release studies were conducted by taking three different amounts of Au NPs (0.50 mg, 0.75 mg and 1.00 mg mask$^{-1}$) and obtained results are given in figure 5. As indicated in figure 5*a*, in the PEO-Gel-AA-COL-GtAu nanofibre system, a maximum release could

be attained even at 10 min (0.75 mg GtAu incorporated system), in contrast to the PEO-Gel-AA-COL-OpAu nanofibre system which requires 20 min or more for a maximum release (figure 5b). However, when 1.0 mg or 0.50 mg of GtAu NPs are added, the release is much more controlled and a highest value (80.3% and 63.9%, respectively) is reached only at or after 20 min of contact with the buffer medium (figure 5a). However, in 0.75 mg of GtAu NP concentration, the highest release of 75.2% was shown at the tenth minute and then gradual reduction of Au release was observed. In PEO-Gel-AA-COL-OpAu added systems (figure 5b), the release from the 1.0 mg incorporated sample is much higher (99.1%) and it reaches a maximum at the twentieth minute, while the 0.75 mg incorporated system shows a similar release pattern by reaching to a maximum level (84.8%) at the thirtieth minute. Nevertheless, the Au NP release from the 0.50 mg PEO-Gel-AA-COL-OpAu system is gradual and similar to the 0.50 mg PEO-Gel-AA-COL-GtAu system owing to the lower loading of Au NPs. Systems with higher amounts of Au NP always exhibit rapid release. Comparing both GtAu NPs and OpAu NPs incorporated PEO-Gel-AA-COL systems, it is much clearer that the PEO-Gel-AA-COL-OpAu system accounts for a higher content of Au release than the GtAu NP system. This indicated that smaller size GtAu NPs could have strongly implanted and interacted with polymer molecules, hindering their release in to the buffer medium in contrast to larger size NPs in the OpAu system, showing weak interaction leading to higher release (figure 5a,b).

## 3.7. Skin permeation of gold nanoparticles released from PEO-Gel-AA-COL nanofibre system

Permeation of Au NPs released from the PEO-Gel-AA-COL nanofibre systems was analysed using Strat membrane which has performances similar to the human skin. As observed in cross-sectional analysis given in the electronic supplementary material, figure S5ai and ii, Strat membrane is consists of three layers that are made out of polyestersulfones [85] that represents the epidermis including stratum corneum, dermis and hypodermis of skin [86]. When the Strat membrane was in contact with PEO-Gel-AA-Col-OpAu nanofibres and PEO-Gel-AA-COL-GtAu nanofibres (electronic supplementary material, figure S5b) it was subjected to EDX analysis, the peaks appeared indicated that Au NPs are distributed in the middle membrane layer with varied intensities without diffusing into the outer layer which is more open and porous (electronic supplementary material, figure S5c,d). This confirmed that the Au NPs produced from both tea tree extract and orange-peel extract were able to diffuse into the epidermis or dermis layer which are generally known to be much resistant (more specifically the stratum corneum) in allowing the permeation of various molecules. Moreover, it was also clearly evinced that these particles are not reaching into the hypodermis or to the blood circulation (absence of an ultraviolet absorbance in the receiver compartment-data not shown).

## 3.8. Radical scavenging activity of PEO-Gel-AA-COL-gold nanoparticle nanofibre systems

According to the electronic supplementary material, figure S6, the results of the DPPH assay revealed that, nanofibres have a much closer activity like the AA standard. More importantly at low concentrations their activity is even higher than the AA (in the range of 1.5–3.0 µg ml$^{-1}$). At higher concentrations the activity of the PEO-Gel-AA-COL-OpAu nanofibre system is much pronounced compared to the respective GtAu nanofibre system (IC$_{50}$ of 26.62 µg ml$^{-1}$ compared to 31.58 µg ml$^{-1}$ of PEO-Gel-AA-COL-GtAu nanofibre system). A possible explanation of the increased antioxidant property of the PEO-Gel-AA-COL-OpAu system could be owing to a myriad of phytochemicals present in the orange peel extract which is much higher in content than in green tea leaf extract [87]. However, in general, the profound radical scavenging activity of the two nanofibre systems could also be attributed to the presence of plant metabolites, and the synergistic effect of Au NPs with AA. [27].

## 3.9. Toxicity assessment of gold nanoparticles released from the facial mask

Toxicity studies of the developed facial mask was carried out using zebra fish embryos. Among tested zebra fish embryos for teratogenicity parameters, yolk sac edema has been significant rather than the other anomalies. The yolk sac is an excellent model for exposing toxicant disruptions to early embryonic nutrition and has the potential to discover mechanistic insights into the developmental origins of health and disease. As depicted in figure 6, zebra fish embryos within the considered duration have not shown any toxic effect against the GtAu nanoparticles for all concentrations. However, yolk sac edema has been observed for OpAu NPs with 208.32 ppm and 166.67 ppm concentrations at 48 hpf and it reveals the comparably higher toxic effect against OpAu NPs at higher

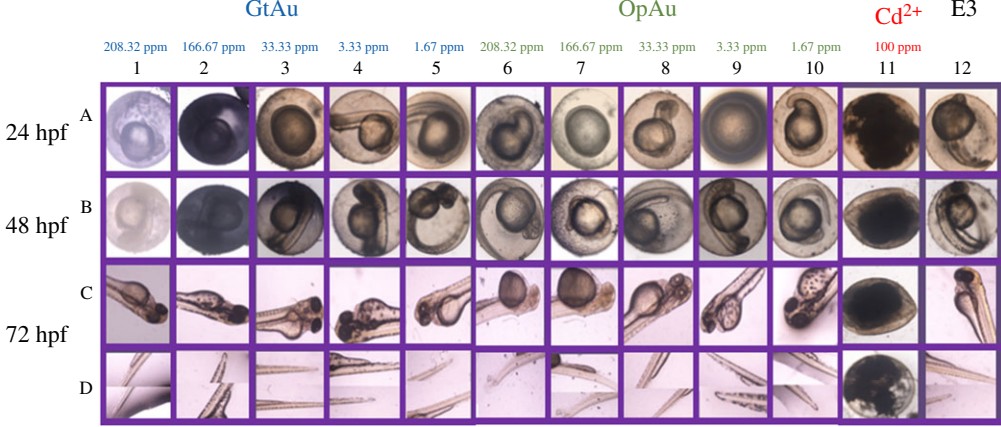

**Figure 6.** Malformations due to Au NP toxicity in developing zebra fish embryos. Zebra fish at 24 hpf (A1–A12), 48 hpf (B1–B12), and 72 hpf (C1–C12) and (D1–D12) were exposed to, E3 (A12, B12, C12, D12), GtAu(A1–A5, B1–B5, C1–C5, D1–D5), OpAu (A6–A10, B6–B10, C6–C10, D6–D10) and Cd$^{2+}$ 100 ppm (A11, B11, C11, D11). Embryos were scored for number of somites including tail detachment, pericardial edema, normal yolk body size and yolk extension. Embryos exposed to OpAu NPs (C6, C7, C8) were less pigmented than untreated embryos. Embryos exposed to 208.32 ppm, 166.67 ppm OpAu NPs had an increased flattened yolk sac with increased width (C6, C7); lateral view. Embryo exposed to Cd$^{2+}$ 100 ppm was unfertilized. Scale bar, 200 µm.

concentrations. This kind of a phenomenon with OpAu could be explained by the shape and size dependent toxicity of Au NPs, as OpAu NPs were creating irregular shapes with the huge aggregation [88]. The positive control, Cd$^{2+}$ at 100 ppm has affected to increase the motility of embryos even after 24 hpf, proving its highest toxicity effect. In the final analysis, it proves that any concentration of GtAu NPs below 208.32 ppm ≤ and lower concentration (166.67 ppm<) of OpAu NPs would not affect the toxicity of zebra fish embryos [89–93].

# 4. Conclusion

In this work, electrospinning technology has enabled the generation of a paper-like thin hydrogel facial mask to deliver Au NPs, AA and COL which could provide anti-ageing, whitening and anti-wrinkling benefits for the user. The presence of corresponding peaks of Au NPs in XRD patterns indicated the successful incorporation of Au NPs in to the electrospun nanofibre systems. SEM results proved that the presence of COL and AA during the preparation of nanofibres is beneficial to entrap the Au NPs into the nanofibre system. DSC results suggested that the shifting of the melting temperature of PEO to lower values is possible when blended with other molecules and formulated into nanofibres, as it alters the crystallinity of the neat PEO. Releasing studies indicated that the PEO-Gel-AA-COL-GtAu nanofibre with 0.75 mg Au NPs is the fastest in releasing entrapped gold at the tenth minute, while a higher content of Au release was observed with the PEO-Gel-COL-OpAu nanofibre system. It was also noticed that the higher the Au content, the much faster the release of Au. This rapid Au release behaviour suggests promising application as a facial mask as it reduces the wearer time below 30 min. Skin permeation studies carried out with Strat membrane indicated the penetration of Au into the dermis and epidermis layers of the skin, highlighting its' potential skin, absorptivity. Similar radical scavenging activity to AA exhibited by PEO-Gel-AA-COL-Au nanofibres further confirmed the antioxidant potential of the facial mask. In vivo studies further demonstrated that selecting Au NPs below the concentration of 208.32 ppm from GtAu NPs and 166.67 ppm from pAu NPs have a promising non-toxic profile on the zebra fish embryo model. Therefore, we can conclude that PEO-Gel-AA-COL nanofibres have promising Au NP releasing activity which could allow it to act as an effective hydrogel mask with proven anti-ageing, whitening, antioxidant and anti-wrinkling benefits.

Ethical. Ethical clearance for animal experiments was approved by the Institute of Biology of Sri Lanka (ERC IOBSL 199 07 2019).

Data accessibility. The datasets supporting this article have been uploaded as part of the electronic supplementary material. In addition, the datasets can also be found at https://doi.org/10.5061/dryad.s1rn8pk5d [94].

Authors' contributions. K.M.N.de.S., R.M.de.S. and D.C.M. conceived the idea and designed the project. D.C.M., V.U.G. and H.M.L.P.B.H. carried out the experiments and D.C.M. wrote the first draft of the paper. C.Y.Y., J.Y.C.,

A.A.A.D.S., S.R. and R.N. contributed immensely to advanced instrumentation analysis. All authors discussed the results and commented on the final manuscript.

Competing interests. Authors have no competing interests.

Funding. This work was supported by National Research Council Sri Lanka (grant no. NRC 18-013).

Acknowledgements. The authors would like to extend sincere gratitude to National Research Council Sri Lanka (NRC18-013) for the financial support provided. The authors are also grateful for Mr Himala Ratnayake and the technical team at Techno Solutions Pvt (Ltd) for instrumentation support in FTIR analysis. Special appreciation to Mr G.V.C.P. Lakshman for zebra fish culturing facilities provided and to Mr Chavin Perera for the technical support given.

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
