## [Reviewer comments · Royal Society Open Science]

Review History

RSOS-201266.R0 (Original submission)

Review form: Reviewer 1 (Yumin Leng)

Is the manuscript scientifically sound in its present form?

Yes

Are the interpretations and conclusions justified by the results?

Yes

Is the language acceptable?

Yes

Do you have any ethical concerns with this paper?

No

Have you any concerns about statistical analyses in this paper?

No

Recommendation?

Accept as is

Comments to the Author(s)

1. The notes in Figure 1 need to be enlarged.
2. The literature needs to be updated, and it may be helpful for the article to be quoted the literature (Microchimica Acta 2019, 186, 803; Langmuir 2017, 33, 6398; Scientific Reports 2016, 6, 28900; Talanta 2015, 139, 8990; Langmuir 2013, 29, 7591) in appropriate places.

Review form: Reviewer 2**Is the manuscript scientifically sound in its present form?**

Yes

Are the interpretations and conclusions justified by the results?

Yes

Is the language acceptable?

Yes

Do you have any ethical concerns with this paper?

No

Have you any concerns about statistical analyses in this paper?

No

Recommendation?

Accept with minor revision (please list in comments)

Comments to the Author(s)

In the conclusion, the authors state that "In-vivo studies on Zebra fish embryo model further demonstrated that these Au NPs have promising non-toxic profile". However, the results in Fig.6 clearly showed that the high concentration of prepared gold nanoparticles could impact on embryo in a negative way. Therefore, it is important to state that the suitable concentration of designed gold nanoparticles is needed to avoid any adverse effect induction.

Decision letter (RSOS-201266.R0)

Dear Professor Nalin De Silva:

Title: Nanofibrous cosmetic face mask for transdermal delivery of nano gold: Synthesis, characterization, release and Zebra fish employed toxicity studies
Manuscript ID: RSOS-201266

Thank you for submitting the above manuscript to Royal Society Open Science. On behalf of the Editors and the Royal Society of Chemistry, I am pleased to inform you that your manuscript will

be accepted for publication in Royal Society Open Science subject to minor revision in accordance with the referee suggestions. Please find the reviewers' comments at the end of this email.

The reviewers and handling editors have recommended publication, but also suggest some minor revisions to your manuscript. Therefore, I invite you to respond to the comments and revise your manuscript.

Because the schedule for publication is very tight, it is a condition of publication that you submit the revised version of your manuscript before 22-Aug-2020. Please note that the revision deadline will expire at 00.00am on this date. If you do not think you will be able to meet this date please let me know immediately.

Kind regards,
Dr Laura Smith
Publishing Editor, Journals

On behalf of the Subject Editor Professor Anthony Stace and the Associate Editor Dr Chaohua Cui.

RSC Associate Editor:
Comments to the Author:
(There are no comments.)

RSC Subject Editor:
Comments to the Author:
(There are no comments.)

Reviewer comments to Author:
Reviewer: 1

Comments to the Author(s)
1. The notes in Figure 1 need to be enlarged.
2. The literature needs to be updated, and it may be helpful for the article to be quoted the literature (Microchimica Acta 2019, 186, 803; Langmuir 2017, 33, 6398; Scientific Reports 2016, 6, 28900; Talanta 2015, 139, 8990; Langmuir 2013, 29, 7591) in appropriate places.

Reviewer: 2

Comments to the Author(s)
In the conclusion, the authors state that "In-vivo studies on Zebra fish embryo model further demonstrated that these Au NPs have promising non-toxic profile". However, the results in Fig.6 clearly showed that the high concentration of prepared gold nanoparticles could impact on embryo in a negative way. Therefore, it is important to state that the suitable concentration of designed gold nanoparticles is needed to avoid any adverse effect induction.

Author's Response to Decision Letter for (RSOS-201266.R0)

See Appendix A.

Decision letter (RSOS-201266.R1)

Dear Professor Nalin De Silva:

Title: Nanofibrous cosmetic face mask for transdermal delivery of nano gold: Synthesis, characterization, release and Zebra fish employed toxicity studies
Manuscript ID: RSOS-201266.R1

It is a pleasure to accept your manuscript in its current form for publication in Royal Society Open Science. The chemistry content of Royal Society Open Science is published in collaboration with the Royal Society of Chemistry.

On behalf of the Subject Editor Professor Anthony Stace and the Associate Editor Dr Chaohua Cui.

RSC Associate Editor
Comments to the Author:
(There are no comments.)

Reviewer(s)' Comments to Author:

Appendix A

We would like to thank both the reviewers. Changes made on the manuscript are highlighted in yellow.

Reviewer	Reviewer comment	Response
Reviewer 1	1. The notes in Figure 1 need to be enlarged2. The literature needs to be updated, and it may be helpful for the article to be quoted the literature (Microchimica Acta 2019, 186, 803; Langmuir 2017, 33, 6398; Scientific Reports 2016, 6, 28900; Talanta 2015, 139, 8990; Langmuir 2013, 29, 7591) in appropriate places	1. Addressed2. Addressed (highlighted in yellow)
Reviewer 2	1. In the conclusion, the authors state that "In-vivo studies on Zebra fish embryo model further demonstrated that these Au NPs have promising non-toxic profile" . However, the results in Fig.6 clearly showed that the high concentration of prepared gold nanoparticles could impact on embryo in a negative way. Therefore, it is important to state that the suitable concentration of designed gold nanoparticles is needed to avoid any adverse effect induction	Addressed and the statement was changed to- “ In-vivo studies further demonstrated that selecting Au NPs below the concentration of 208.32 ppm from GtAu NPs and 166.67 ppm from pAu NPs have a promising non-toxic profile on Zebra fish embryo model”. (Highlighted in yellow)